# Association of small, dense LDL-cholesterol concentration and lipoprotein particle characteristics with coronary heart disease: A systematic review and meta-analysis

Lathan Liou◉*, Stephen Kaptoge

Department of Public Health and Primary Care, University of Cambridge, Cambridge, United Kingdom

* ll587@cam.ac.uk

## Abstract

### Objectives

The aim of this study was to systematically collate and appraise the available evidence regarding the associations between small, dense low-density lipoprotein (sdLDL) and incident coronary heart disease (CHD), focusing on cholesterol concentration (sdLDL-C) and sdLDL particle characteristics (presence, density, and size).

### Background

Coronary heart disease (CHD) is the leading cause of death worldwide. Small, dense low-density lipoprotein (sdLDL) has been hypothesized to induce atherosclerosis and subsequent coronary heart disease (CHD). However, the etiological relevance of lipoprotein particle size (sdLDL) versus cholesterol content (sdLDL-C) remains unclear.

### Methods

PubMed, MEDLINE, Web of Science, and EMBASE were systematically searched for studies published before February 2020. CHD associations were based on quartile comparisons in eight studies of sdLDL-C and were based on binary categorization in fourteen studies of sdLDL particle size. Reported hazards ratios (HR) and odds ratios (OR) with 95% confidence interval (CI) were standardized and pooled using a random-effects meta-analysis model.

### Results

Data were collated from 21 studies with a total of 30,628 subjects and 5,693 incident CHD events. The average age was 67 years, and 53% were men. Higher sdLDL and sdLDL-C levels were both significantly associated with higher risk of CHD. The pooled estimate for the high vs. low categorization of sdLDL was 1.36 (95% CI: 1.21, 1.52) and 1.07 (95% CI: 1.01, 1.12) for comparing the top quartiles versus the bottom of sdLDL-C. Several studies suggested a dose response relationship.

**Data Availability Statement:** All relevant data are within the manuscript and its Supporting Information files.

**Funding:** The author(s) received no specific funding for this work.

## Conclusions

The findings show a positive association between sdLDL or sdLDL-C levels and CHD, which is supported by an increasing body of genetic evidence in favor of its causality as an etiological risk factor. Thus, the results support sdLDL and sdLDL-C as a risk marker, but further research is required to establish sdLDL or sdLDL-C as a potential therapeutic marker for incident CHD risk reduction.

## Introduction

Coronary heart disease (CHD) is the leading cause of death worldwide, with an estimated 7.4 million people having died from CHD in 2015 [1]. In the USA, although mortality rate has been decreasing, the prevalence of CHD is predicted to rise from 6.8% (2015) to 8.2% (2035) [2]. With the projected increased burden, it is increasingly important to identify risk factors that can help to identify high CHD-risk individuals. CHD is primarily caused by atherosclerosis and the resulting inflammation of the coronary arteries [3]. Although low-density lipoprotein cholesterol (LDL-C) is a well-studied risk factor, there is a growing body of evidence that challenges the conventional view of LDL-C as the most relevant biomarker for CHD. Firstly, individuals with normal range LDL-C have been found to still develop CHD [4] and secondly, several observational studies have found that adjusting for other lipoproteins substantially attenuates the association of LDL-C [5–7], which suggests that other novel lipoproteins may have more discriminatory potential.

Circulating lipoproteins vary in size, density, and composition, and various laboratory methods have been developed to separate LDL fractions into subfractions. The first method, ultracentrifugation, separated LDL particles based on flotation rate into generally four subclasses, LDL I (density = 1.025–1.034 g/ml), II (1.034–1.044 g/ml), III (1.044–1.060 g/ml), and IV (>1.060 g/ml) where LDL I and II characterizes phenotype A (large buoyant LDL), and LDL III and IV characterizes phenotype B (small, dense LDL) [8]. Another analytical method is gradient gel electrophoresis (GGE) under nondenaturing conditions, which separates LDL particles by their size and shape. Studies using GGE define four subclasses as well LDL I (large LDL, peak diameter 26.0–28.5 nm), LDL II (intermediate LDL, 25.5.-26.4 nm), LDL III A and B (small LDL, 24.2–25.5 nm), and LDL IV A and B (very small LDL, 22.0–24.1 nm) [9]. There is a strong correlation between density and size of particles analyzed by ultracentrifugation and GGE respectively. Other methods include NMR which subclassifies LDL particles based on size and automated homogeneous assays, which separates sdLDL fractions with a density from 1.044 to 1.063 g/ml [10,11].

Small, dense low-density lipoproteins (sdLDL) have been increasingly studied as a better marker for cardiovascular disease outcomes. They were initially described by Krauss to be associated with relative increases in plasma triglyceride and apolipoprotein B and posited to potentially underlie a familial predisposition to CHD [12]. Austin has produced a large body of research further linking triglycerides and sdLDL [13] as well as positing sdLDL as a risk factor for CHD, albeit based only on case control and cross-sectional studies [14]. The number of sdLDL particles was reported to be a more sensitive biomarker for metabolic syndrome compared to LDL-C [15], and sdLDL-cholesterol (sdLDL-C), the free cholesterol content within sdLDL particles, was reported to be a better marker for assessment of CHD than total LDL-C [16]. Moreover, sdLDL is currently accepted as a risk factor for CVD by the National Cholesterol Education Program [17]. While there is high validity between sdLDL particle

measurement analyzed by ultracentrifugation and gel electrophoresis [18], the agreement between these conventional methods and nuclear magnetic resonance is yet to be validated. Its physical and biochemical properties have been hypothesized and widely believed to facilitate its atherogenic potential.

The origins of sdLDL formation are hypothesized to be from the delipidation of triglyceride-rich lipoproteins catalyzed by lipoprotein lipase and hepatic lipase enzymes [19], and sdLDL has been associated with elevated plasma triglyceride levels, reduced HDL cholesterol, and high hepatic lipase activity [20]. In fact, evidence of the metabolic role that increased plasma triglyceride levels has on circulating sdLDL levels has been elucidated [21]. The small size of sdLDL particles favors their penetration into the arterial wall where they can instigate cholesterol accumulation and their susceptibility to oxidation attracts inflammatory factors which increase the probability of atherogenesis [22,23]. Further, the circulation time of sdLDL is longer than that of LDL particles, which suggests that there are more opportunities for sdLDL to play an important role in the development and growth of atherosclerotic plaques [24,25].

Research has focused on studying either the levels of sdLDL particle concentrations or the levels of cholesterol within sdLDL particles (sdLDL-C); however only one study has simultaneously looked at both. They found that elevated sdLDL-C concentration, but not sdLDL particle concentration, was found to be a significant marker of CHD risk [26]. The aim of this study was to systematically review and critically appraise existing evidence and quantify both the associations between sdLDL particle concentration and CHD and sdLDL-C concentration and CHD. The findings should provide a comparison of the potential importance of sdLDL versus sdLDL-C as etiological biomarkers for primary occurrence of CHD.

## Methods

### Data source and search

PubMed, MEDLINE (1946 to January 29, 2020), EMBASE (1974 to January 29, 2020), and Web of Science were searched using the search terms for sdLDL, sdLDL-C, CHD, and the measure of association presented in **S1 Table in S1 File**. Literature searches were limited to English-language primary research publications in humans. The searches were supplemented by screening reference lists of included studies and selected reviews. The search was conducted by one investigator (LL). A review protocol does not exist.

### Study eligibility criteria

Titles and abstracts were screened, and available English full texts were retrieved and examined for inclusion. Any studies of a prospective or case control design which reported a measure of association between sdLDL or sdLDL-C and incident CHD with serum (or plasma) samples obtained before determination of outcomes were included. Prospective studies of people with cardiovascular disease at baseline were excluded as first incidence of CHD was the outcome of interest. Prospective studies and case control studies that investigated populations that had other established diseases like diabetes or HIV were included provided there was no evidence of previous cardiovascular disease. The outcome of interest, incidence of CHD, was defined according to ICD10 codes I20-I25: a group of diseases that includes stable angina, unstable angina, myocardial infarction, death due to any of the aforementioned cardiac events, and sudden coronary death [27]. The Grading of Recommendations Assessment, Development and Evaluation (GRADE) tool [28] was used to assess risk of bias in each study (**S2 Table in S1 File**).

## Measurement of exposure

In this review, studies that quantified either the concentration of cholesterol within sdLDL particles (sdLDL-C) or the presence or concentration of sdLDL particles were included. Presence was defined either as LDL classes III (1.044–1.060 g/ml), and IV (>1.060 g/ml) for studies that used ultracentrifugation, LDL classes III A and B (small LDL, 24.2–25.5 nm) for studies that used GGE, or fractions with a density from 1.044 to 1.063 g/ml for studies that used homogenous assay methods.

## Data extraction

Data were collated on population type (general vs. high-risk populations); mean age; sex; geographical location; study design; hypertension prevalence; diabetes prevalence; sample type (serum vs. plasma); assay type; number of participants; and number of incident CHD events. Detailed information about the study setting and the definition of CHD were also collected. Measures of association (odds ratios and hazard ratios) between sdLDL or sdLDL-C and incident CHD were extracted with the following levels of covariate adjustment, when available: 1) unadjusted, adjusted only for age and sex, or vague specification of adjusted covariates; 2) adjusted for demographic factors and conventional CHD risk factors; and 3) adjusted for demographic factors, conventional CHD risk factors, and other lipid levels. The measures of association were standardized (**S3 Table in S1 File**).

## Statistical analysis

All statistical analyses and visualization were performed in R 4.0 using "meta" [29] and "ggplot2" [30], and a 2-sided p-value of <0.05 represented statistical significance. Original measures of association and methods of standardization are reported in **S3 Table in S1 File**. The assessment of publication bias was assessed graphically with funnel plots and Egger's test. For the primary analysis, adjusted odds ratios and hazard ratios for the association between sdLDL-C and CHD were pooled by random effects inverse-variance weighted random effects meta-analysis [31]. The random effects method was selected a priori due to anticipated heterogeneity in the populations studied and the design of included studies. The presence of between study heterogeneity was assessed using the $I^2$ statistic, a statistic that quantifies the percentage of the total observed heterogeneity that is due to between-study variation. The Cochrane Handbook suggests that an $I^2$ between 30–60% may represent moderate heterogeneity and an $I^2$ between 50–90% may represent substantial heterogeneity [32].

Secondary analyses included random effects meta analyses subgrouping by adjustment level (unadjusted and adjusted measures included) and study design. Potential explanatory covariates such as location, population type, and assay method were explored as factors for heterogeneity using univariate meta-regression after adjusting for study design. One study provided mean cholesterol concentration for each quartile, whereas two other studies provided the overall mean and standard deviation of cholesterol concentration. By assuming a normal distribution, the mean cholesterol concentration for each quartile were calculated. A conversion factor of 0.02586 was used to convert from concentrations reported in mmol/L to mg/dL [1]. The mean concentrations were plotted against the corresponding quartile hazard ratios for all three studies. Dose-dependency between sdLDL levels and CHD could not be assessed due to lack of data.

SK was supported by the British Heart Foundation (BHF) (RG/18/13/33946). LL did not receive funding.

## Results

The literature search yielded 1,384 total citations, of which 858 were screened and ultimately 21 met eligibility criteria (**Fig 1**), reporting data on 30,628 participants (53.1% male) with 5,693 CHD events. Thirteen studies used sdLDL as their exposure (hereby referred to as sdLDL studies), whereas the other seven used sdLDL-C (hereby referred to as sdLDL-C studies) and one used both. Nine studies were case control studies, five were nested case-control studies, two were randomized controlled trials, and the remaining five were prospective cohort studies. Studies were conducted in East Asia (n = 7), Europe (n = 6), and North America (n = 8). Seven studies used a sample of the general population, and fourteen studied either hospital participants, participants with type 2 diabetes, HIV, or liver disease. Ten studies measured sdLDL-C/sdLDL using an automated chemical analyzer, seven with gel electrophoresis, and four with nuclear magnetic resonance imaging (NMR). The included studies are summarized in **Table 1**. More detailed descriptions of the study population, outcome definitions, covariate adjustment are provided in **S4 Table in S1 File**. Further details of the sdLDL-C assays including assessments of validity and limitations are reported in **S5 Table in S1 File**.

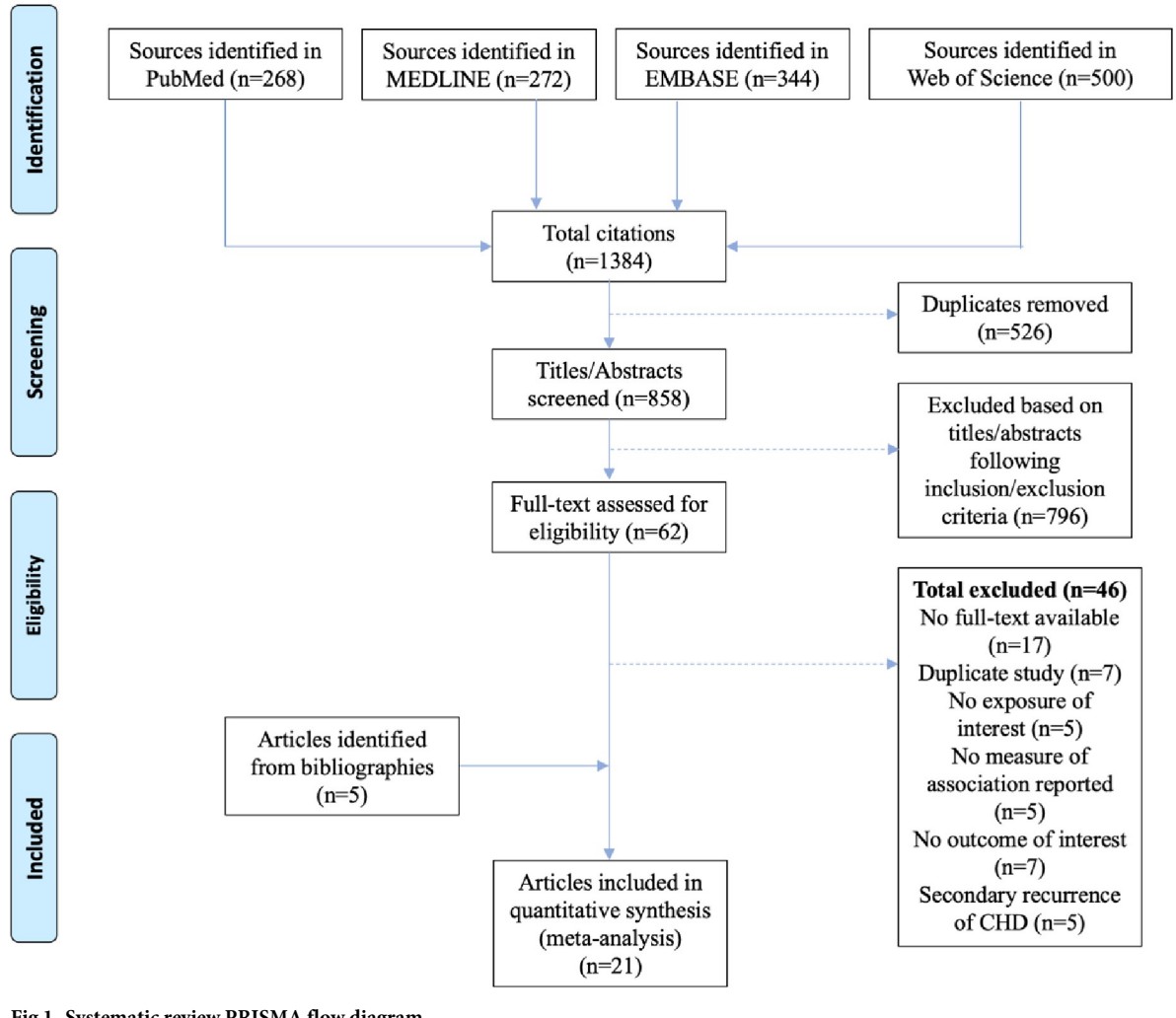

**Fig 1. Systematic review PRISMA flow diagram.**

**Table 1. Summary of studies of association between sdLDL/sdLDL-C and CHD.**

| Author/Date | Study | Study Design | Population | Location | sdLDL-C Assay Method | Mean Age | Sample Size (Cases) | Males |
|---|---|---|---|---|---|---|---|---|
| sdLDL | | | | | | | | |
| Koba et. al 2002 A[†][55] | Showa | CC | Diabetes | East Asia | electrophoresis | 60.3 | 348 (130) | 348 |
| Koba et. al. 2002 B[†] [33] | Showa | CC | Diabetes | East Asia | electrophoresis | 60.2 | 874 (571) | 597 |
| Bucher et. al. 2012 [6] | SHCS | NCC | HIV | Europe | analyzer | NR | 490 (98) | 385 |
| Goliasch et. al. 2011 [56] | MIVYA | CC | Hospital | Europe | electrophoresis | 37.3* | 302 (92) | 263 |
| Kuller et. al. 2002 [57] | CHS | CC | Hospital | North America | NMR | 73 | 373 (191) | 0 |
| Kwon et. al. 2006 [58] | Yonsei | CC | Hospital | East Asia | electrophoresis | 60.4 | 504 (262) | NR |
| Mackey et. al. 2015 [59] | WHI-OS | NCC | Hospital | North America | NMR | 65.1 | 677 (124) | 0 |
| Mykannen et. al. 1999 [60] | Kuopio | NCC | Diabetes | Europe | electrophoresis | 69.2 | 258 (86) | 129 |
| Lamarche et. al. 1997 [61] | QC | NCC | General | North America | electrophoresis | 58 | 2103 (113) | 2103 |
| Otvos et. al. 2006 [39] | VA-HIT | NCC | Veterans | North America | NMR | 64.2 | 1061 (364) | 1061 |
| Russo et. al. 2014 [34] | Messina | CC | Diabetes | Europe | analyzer | 65.3 | 95 (59) | 0 |
| Williams et. al. (2013) [38] | HATS | RCT | Hospital | North America | electrophoresis | 53.6 | 142 (142) | 125 |
| Xu et. al. 2015[‡] [62] | FuWai | CC | Hospital | East Asia | electrophoresis | 55.1 | 413 (293) | 254 |
| Zeljkovic et. al. 2008 [63] | ICDCC | CC | Hospital | Europe | electrophoresis | 55.7 | 359 (181) | 216 |
| sdLDL-C | | | | | | | | |
| Arai et. al. 2013 [5] | Suita | PC | General | East Asia | analyzer | 58.5 | 2034 (63) | 968 |
| Arsenault et. al. 2007 [64] | EPIC-Norfolk | NCC | General | Europe | electrophoresis | 65.7 | 2955 (1035) | 1869 |
| Higashioka et. al. 2019 [1] | Hisayama | PC | General | East Asia | analyzer | 63.2 | 3080 (79) | 1290 |
| Hoogeveen et. al. 2015 [46] | ARIC | PC | General | North America | analyzer | 62.8 | 10225 (1158) | 4499 |
| Siddiqui et. al. 2019 [65] | CRLTR | PC | LTR | North America | NMR | 58 | 130 (20) | 81 |
| Koba et. al. 2008 [7] | Showa | CC | Hospital | East Asia | analyzer | 60.5 | 871 (482) | 612 |
| Tsai et. al. 2014 [26] | MESA | PC | General | North America | analyzer | 60.7 | 3334 (150) | 1473 |

*Only reported median

[†]A refers to *Atherosclerosis* study; B refers to *Am Heart J* study

[‡]Xu et. al. reported both sdLDL and sdLDL-C estimates **Abbreviations:** PC = prospective cohort study; NCC = nested case-control study; CC = case-control study; LTR = liver transplant recipients; NR = not reported; NMR = nuclear magnetic resonance imaging; analyzer = automated chemical analyzer; electrophoresis = gel electrophoresis **Full Study Names:** Suita = Suita Study, EPIC-Norfolk = European Prospective Investigation into Cancer in Norfolk Prospective Population Study; SHCS = Swiss HIV Cohort Study; MIVYA = Myocardial Infarction Survivors in Very Young Adults Study; CHS = Cardiovascular Health Study; Hisayama = Hisayama Study; ARIC = Atherosclerosis Risk in Communities Study; Showa = Showa Study; QBC = Quebec Cardiovascular Study; VA-HIT = Veterans Affairs High-Density Lipoprotein Intervention Trial; Messina = Messina Study; HATS = HDL-Atherosclerosis Treatment Study; MESA = Multi-Ethnic Study of Atherosclerosis; Yonsei = Yonsei Study; WHI-OS = The Women's Health Initiative Observational Study; Kuopio = Kuopio Study; CRLTR = Cardiometabolic Risk in Liver Transplant Recipients Study; ICDCC = Institute of Cardiovascular Diseases, Clinical Centre of Serbia Study

## Association of sdLDL/sdLDL-C with incident CHD

The random effects pooled OR for high vs low sdLDL (14 studies) was 1.36 (95% CI: 1.21, 1.52), with high heterogeneity $I^2$ = 89% (**Fig 2A**). The relative risk for CHD comparing the top versus the bottom quartiles of sdLDL-C (8 studies) was 1.07 (95% CI: 1.01, 1.12), with evidence of substantial heterogeneity ($I^2$ = 87%) (**Fig 2B**).

## Association of sdLDL/sdLDL-C with incident CHD across subgroups

Subgroup analyses by adjustment level and study design were planned a priori. Study-specific unadjusted and adjusted measures of association were first compared in a forest plot sub-grouped by adjustment level (**S1 Fig in S1 File**). For the sdLDL studies, adjustment attenuated the ORs with the exception of two studies [33,34]; however, these two studies adjusted for a limited number of confounders. For the sdLDL-C studies, adjustment to any degree attenuated

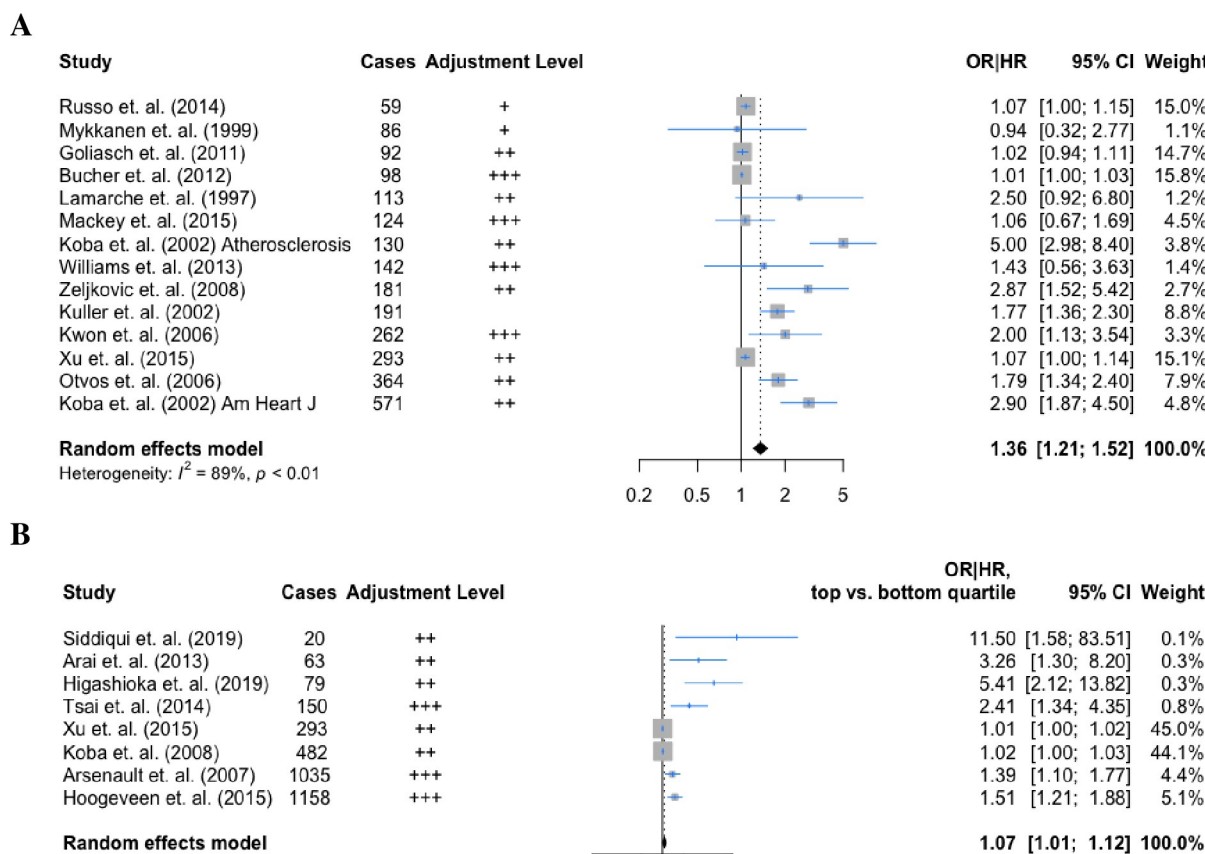

**Fig 2. Forest plots for random effects meta-analysis of the associations between (A) sdLDL, (B) sdLDL-C and CHD.** '+' = adjusted for other lipid subfractions; '++' = adjusted for demographics and lifestyle risk factors; '+++' = adjusted for demographics, lifestyle risk factors and lipid subfractions. Maximally adjusted hazard ratios from each study were used. Kuller et. al. provided an unadjusted estimate only.

the OR to the null (range of % attenuation: 2.3–114.9), although there was no evidence the adjustment extent was associated with the magnitude of attenuation.

The pooled hazard ratio for the nested case-control subgroup (4 studies) provided no evidence of association; 1.03 (95% CI: 0.91, 1.16), although the heterogeneity was low ($I^2$ = 7%). The pooled hazard ratio for the case-control subgroup (8 studies) provided evidence of an association; 1.55 (95% CI: 1.29, 1.86), with substantial heterogeneity ($I^2$ = 91%) (**Fig 3A**). The pooled hazard ratio for the randomized controlled trial subgroup (2 studies) provided evidence of an association; 1.76 (95% CI: 1.33, 2.32), with no heterogeneity. There was evidence of publication bias (Egger's test: 4.4, p-value = $7.9 \times 10^{-4}$) (**S2A Fig in S1 File**) The pooled hazard ratio for the prospective cohort subgroup in sdLDL-C studies (5 studies) was 2.83 (95% CI: 1.57, 5.09) and 1.01 (95% CI: 0.99, 1.04) for the case-control subgroup (Arsenault et. al. was classified as a case-control) (3 studies). The heterogeneity lowered from 85% to 71% for the prospective cohort subgroup (**Fig 3B**). There was evidence of publication bias (Egger's test: 14.8, p = $6.1 \times 10^{-6}$) (**S2B Fig in S1 File**).

Univariate meta-regressions for location, assay method, population type and publication year were performed after adjusting for study design for studies reporting sdLDL (**S6 Table in S1 File**) and sdLDL-C (**S7 Table in S1 File**). The results showed that sdLDL studies using European populations and sdLDL-C studies using unhealthy populations or gel

electrophoresis methods tended to report lower measures of association. sdLDL-C studies using North American populations tended to report slightly higher measures of association. Interestingly, location and population type individually appeared to explain 56% and 72% of the variability in reported measures of association in sdLDL-C studies respectively.

### Assessment of a dose-response relationship

In each of three studies that provided information on quartiles of sdLDL-C, there was a significant trend association found across quartiles, providing evidence of a dose-dependent relationship between sdLDL-C and risk of CHD (**Fig 4**).

## Discussion

This systematic review and meta-analysis of adjusted association estimates from 19 observational studies and 2 randomized suggests that the presence of sdLDL is associated with increased risk of developing CHD. This association was independent of conventional cardiovascular risk factors and other lipid subfractions, as well as consistent across different measurement methods. Furthermore, there was some evidence of a dose-response relationship with sdLDL-C concentration, albeit heterogeneous across studies. The pooled association between sdLDL-C and incident CHD was also independent of conventional cardiovascular risk factors and other lipid subfractions, as well as consistent across geographical regions. Interestingly, when subgrouped by adjustment level, the sdLDL studies that had adjusted for LDL-C (an established lipid biomarker), had a non-statistically significant pooled estimate of 1.54 (95% CI: 0.97, 2.43). However, this could be due to chance in the 21 studies included (5,693 CHD cases) and requires more powerful analyses to more conclusively assess whether sdLDL has no prognostic value in addition to established lipid biomarkers. The pooled estimate for CHD risk in prospective cohort studies was 2.83 (95% CI: 1.57, 5.09). In addition, sdLDL does seem to have high discriminative potential as several studies reported area under the curves (AUC) between sdLDL and CHD as low as 0.641 [35] and as high as 0.83 [36] in a Chinese and Indian cohort respectively, which suggests that sdLDL has a high sensitivity and specificity of predicting CHD. sdLDL also has a fairly high AUC, 0.74, in diabetic participants [37].

The strongest evidence for the potential of sdLDL as a biomarker come from randomized controlled trial results. Williams et. al. not only found that lower sdLDL was independently associated with lower risk of CHD in the HATS trial, but also that simvastin and niacin significantly reduced the levels of sdLDL by 29% (p-value = 0.002) [38]. In addition other trials such as VA-HIT [39] and a trial conducted in hypercholesterolemia patients [40] showed that other treatments besides statins (gemfibrozil, a fibric acid derivative, and mipomersen, an apoB inhibitor) predominantly reduced the concentration of sdLDL particles. Taken together, this suggests that sdLDL may be a potential target for lipid-lowering interventions. While the American Heart Association currently recommends treatment based on a patient's LDL-C levels (> 160 mg/dL) [41], given the existing body of evidence from RCTs, a similar guideline could be suggested for sdLDL. The European Society of Cardiology (ESC) and European Atherosclerosis Society (EAS) have characterized a pattern of dyslipidemias (termed the atherogenic lipid triad) which predispose premature CVD, characterized by increased LDL/triglyceride levels, increased sdLDL levels, and reduced high-density lipoprotein-cholesterol levels (Level B evidence, Class IIa recommendation) [42]. Existing data on the association between sdLDL and CHD come from mostly large non-randomized studies and two recent RCTs, and thus the weight of the evidence in favor of treatment based on sdLDL should be considered. Further evidence on the causal association between sdLDL-C and CHD is motivated, given that there only exist randomized controlled trials studying sdLDL. However,

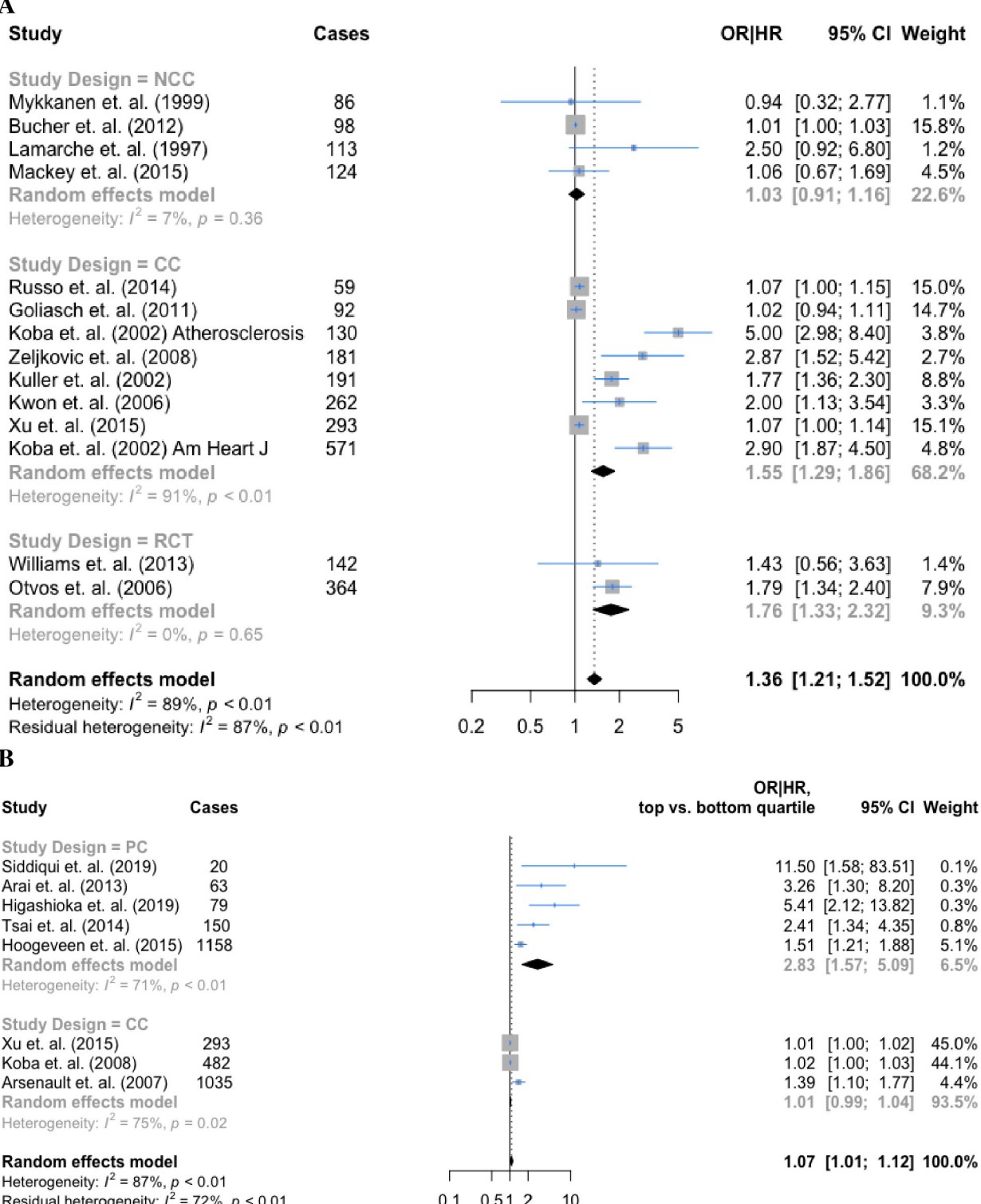

**Fig 3.** Forest plots of association between (A) sdLDL, (B) sdLDL-C and CHD subgrouped by study design. PC = prospective cohort study; CC = case-control study; NCC = nested case-control study; RCT = randomized controlled trial.

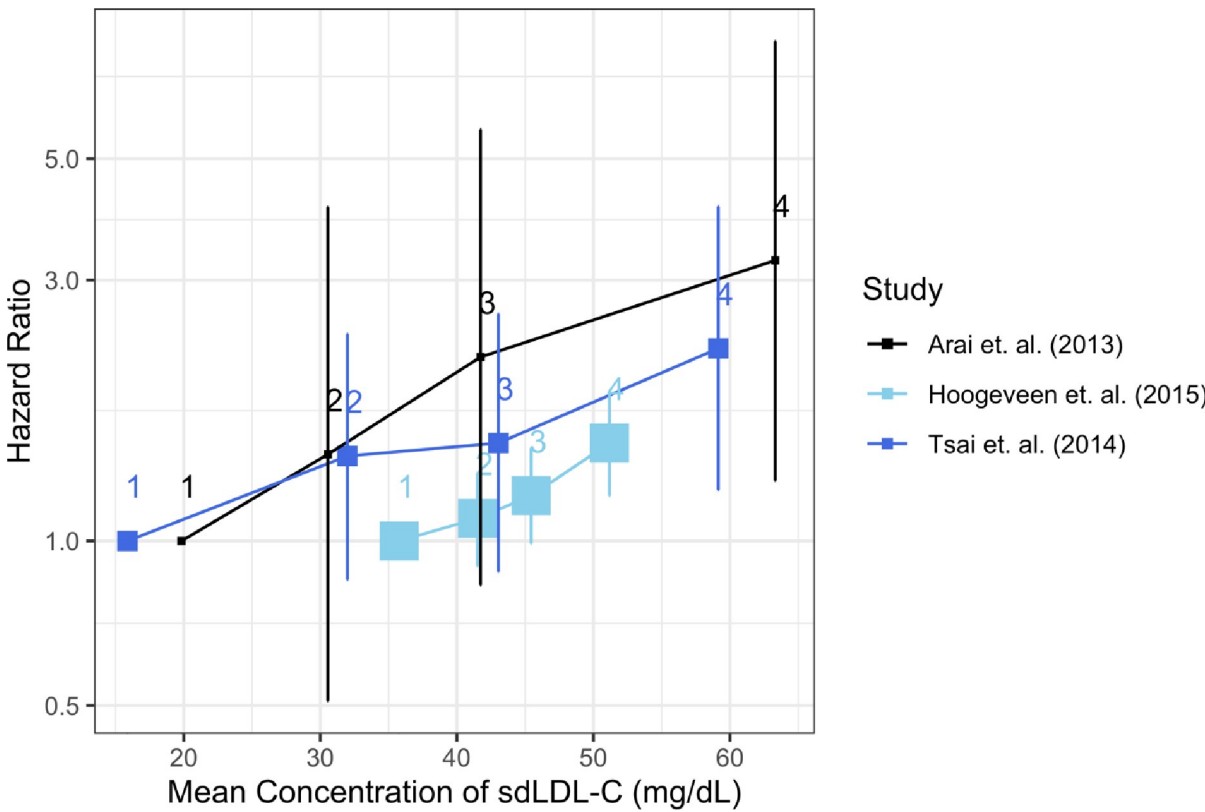

**Fig 4. Dose-dependent relationships in three studies that report quartiles of sdLDL-C.** The numbers 1–4 represent the quartiles. The quartiles used by Arai are ≤19.8, 19.8–30.6, 30.6–41.7, 41.7–63.3. The calculated quartiles for Hoogeveen are ≤35.8, 35.8–41.5, 41.5–45.4, 45.4–51.2 and for Tsai, ≤15.9, 15.9–32.0, 32.0–43.0, 43.0–59. The size of the square represents the sample size, with Hoogeveen having the largest sample size (n = 10,225).

considering the biological evidence, extant observational evidence, extant RCT evidence and an increasing amount of genetic evidence, studying both sdLDL and sdLDL-C-lowering interventions on CHD risk to investigate their biomarker potential is indicated.

Mendelian Randomization (MR) is a widely agreed upon approach to establish temporal causal evidence due to the random allocation of alleles and their precedence before potential confounders in a lifetime [43]. Previously, GWAS analysis identified SNPs clustered at 8 different loci on chromosomes 1, 2, 7, 8, 11 and 19 that were significantly associated with sdLDL-C [44]. For instance, one such SNP, rs4420638, was found to be associated with lipoprotein-associated phospholipase A2, which generates proinflammatory and proatherogenic compounds in the arterial wall and is considered a potential therapeutic target for CHD [45]. Furthermore, Hoogeveen et. al. found the SNP rs508487 located in the *PCSK7* gene (implicated in internalization of LDL receptors) to be significantly associated with sdLDL-C at the genome-wide level in a GWAS of the Atherosclerosis Risk in Communities cohort [46]. It is plausible that PCSK7 is involved with modulating circulating lipid levels and may hold promise as another therapeutic target for CHD. Zhao et. al. found that the SNPs for sdLDL (OR: 1.45; p-value = 0.043) and sdLDL-C (OR: 1.43; p-value = 0.042) were significant predictors of CHD in their multivariable MR analysis tested in the CARDIOgGRAMplusC4D and UK Biobank datasets adjusted for HDL, LDL-C and triglycerides, although they do not specify which SNPs [47]. Further, they did not find statistical evidence for horizontal pleiotropy for the SNPs associated with sdLDL, which lends credence to the validity assumption of those SNPs as instrumental variables.

Ference et. al. also demonstrated via MR that triglyceride-lowering lipoprotein lipase variants and low-density lipoprotein cholesterol-lowering variants were associated with lower risk of CHD per 10-mg/dL of apolipoprotein B-containing lipoproteins [48]. Given the hypothesized mechanism between lipase and sdLDL formation, it would be informative to similarly assess the association between sdLDL-lowering variants and risk of CHD.

sdLDL and sdLDL-C may be useful as biomarkers to identify high-risk individuals and allow for early prevention as studied for LDL-C via a combination of diet and exercise [49] and statin therapy [50], although such an approach would require further investigation. Specifically, with diet, an emerging field of research aims to understand the metabolomics of lipoproteins following a postprandial lipemia response to a meal. Standard fasting conditions prior to bloodwork are not necessarily representative of normal lifestyles. Although it has been hypothesized that hepatic lipase activity increases postprandially and results in the formation of more sdLDL particles, several studies have not been able to demonstrate a statistically significant increase in sdLDL levels [51,52]. Clinically, sdLDL has already gained some recognition as a potential biomarker by the National Cholesterol Education Program (NCEPIII) [17]. The mechanisms triggering the release of sdLDL in people without clinically manifest CHD still remain unclear. It is speculated that low plasma triglyceride levels in participants with familiar hypercholesterolemia [53] or hypertriglyceridemia [21] may affect apolipoprotein metabolic regulatory networks which promote elevated levels of circulating sdLDL.

## Study strengths and limitations

This study is the first meta-analysis of available evidence from observational studies investigating the association between sdLDL/sdLDL-C and CHD using standardized measures of association to allow for comparison. We pooled data from 21 studies with a total of 30,628 subjects and 5,693 incident CHD events, providing substantial statistical power.

There were a number of limitations that warrant discussion. Even though the heterogeneity was fairly high amongst both sdLDL and sdLDL-C studies, the random effects meta-analysis was conducted and interpreted because there was no inconsistency in the *direction* of the effect, but rather its magnitude. Misclassification bias may occur to a different extent across studies as CHD diagnosis has likely changed over time and may vary from doctor to doctor, which may contribute to the observed between-study heterogeneity. Bias from measurement error may occur since sdLDL and sdLDL-C were measured using different methods, and certain studies only performed one measurement [1,5], which may result in OR's attenuated towards the null. The funnel plots and Egger's tests suggested some evidence of publication bias present within the studies with smaller sized studies tending to report more extreme estimates. Adjustment for confounding was not to the same extent in all studies, which may contribute to heterogeneity (**S1 Fig in S1 File**). It was difficult to judge across studies whether adjustment attenuated the association towards the null overall as sample sizes in unadjusted and adjusted models often differed within studies. Furthermore, some studies may have adjusted for mediators (namely other lipids whose position on the causal pathway is unclear), potentially leading to vastly attenuated association estimates. One potential residual confounder was socioeconomic status, which has been widely studied as risk factor for CHD [54]. Further, all included studies were conducted in North America, Europe, or East Asia which limited generalizability. The extent of validity of the assumption of a normally distributed exposure variable and a linear association with the outcome of interest could only be approximately inferred and generalized from few available study-specific evidence, including only one study reviewed that reported a normal distribution of sdLDL-C and approximate dose-response plots constructed from reported estimates. Thus, this study is unable to determine

whether there exists a dose-dependent risk or a threshold risk based on the concentration of sdLDL-C or sdLDL particles. Different studies used different quartile cutoffs potentially limiting comparability between these studies that may be more concerning if the exposure distributions differ greatly across studies. Important limitations of each study reviewed are listed in **S8 Table in S1 File**. Finally, we did not include studies recording other cardiovascular outcomes (i.e. non-CHD), which limits the conclusions drawn from this study on the prognostic value of sdLDL and sdLDL-C to primary prevention of CHD.

## Conclusions

Both sdLDL and sdLDL-C are associated with higher CHD risk. The results are concordant with research investigating related lipids and is supported by biological evidence of sdLDL's atherogenic potential, dose response evidence as well as genetic association studies. The implications are that while sdLDL/sdLDL-C may be useful as a risk marker, further research needs to be done to assess whether it is a suitable therapeutic target independent of well-known lipid metabolism pathways that have proven target therapies. Future research should aim to better characterize the dose-dependency between sdLDL levels and CHD, which could not be assessed due to lack of detailed information and investigate whether the simultaneous determination of sdLDL and sdLDL-C concentrations improve prognosis of CHD risk.

## Supporting information

**S1 File.**
(DOCX)

## Author Contributions

**Conceptualization:** Lathan Liou.

**Data curation:** Lathan Liou.

**Formal analysis:** Lathan Liou.

**Investigation:** Lathan Liou.

**Methodology:** Lathan Liou.

**Supervision:** Stephen Kaptoge.

**Validation:** Stephen Kaptoge.

**Visualization:** Lathan Liou.

**Writing – original draft:** Lathan Liou.

**Writing – review & editing:** Stephen Kaptoge.

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
