## [Decision Letter · Decision Letter 0]

17 Jul 2020

PONE-D-20-14231

Small, Dense LDL-Cholesterol and Coronary Heart Disease: A Systematic Review and Meta-analysis

PLOS ONE

Dear Dr. Liou,

Thank you for submitting your manuscript to PLOS ONE. After careful consideration, we feel that it has merit but does not fully meet PLOS ONE’s publication criteria as it currently stands. Therefore, we invite you to submit a revised version of the manuscript that addresses the points raised during the review process in a point by point fashion.

We look forward to receiving your revised manuscript.

Kind regards,

Andreas Zirlik, MD

Academic Editor

PLOS ONE

Journal Requirements:

2. Please complete a quality assessment of individual studies included in the meta-analysis and report the methodology used in the methods section (items #12 and #19 in the PRISMA checklist).

Reviewers' comments:

Reviewer's Responses to Questions

**Comments to the Author**

1. Is the manuscript technically sound, and do the data support the conclusions?

Reviewer #1: Yes

Reviewer #2: Partly

2. Has the statistical analysis been performed appropriately and rigorously? 

Reviewer #1: Yes

Reviewer #2: Yes

3. Have the authors made all data underlying the findings in their manuscript fully available?

Reviewer #1: Yes

Reviewer #2: Yes

4. Is the manuscript presented in an intelligible fashion and written in standard English?

Reviewer #1: Yes

Reviewer #2: Yes

5. Review Comments to the Author

Reviewer #1: Review PVZ

GENERAL COMMENTS

This work is a meta-analysis paired with a narrative review on data regarding small dense (sd) LDL and their cholesterol content. A total of 17 observational studies including about 29000 subjects who suffered 4900 events (i.e. around 17%) entered the meta-analysis. The authors investigate the predictive value of these parameters for the incidence of coronary artery disease. From this background, it is necessary

1) to more clearly state what is sdLDL(-C);

2) which particle characteristics are measured thereby; and

3) to more clearly state that the reports included investigated healthy of high risk individuals (which they do) but not patients with established disease (which they do not)

SPECIFIC COMMENTS

The Introduction is appropriate to introduce the topic to the readership of an open access journal with holistic spectrum. However, one misses quotes on the pioneering work of Melissa Austin and the connection to triglyceride handling by Ron Krauss.

Methods

Again, it appears necessary to give more details on measurement of sd particles, also on their total cholesterol content, free and esterified cholesterol composition. Supplementary Table 4 is not sufficient for the reader who should learn about the validity and limitations of the assays in clinical terms

Results

These are clearly stated. However, one misses more details on the absolute values of sd particle and their prospective value. One would like to know if there is a continuous risk or a threshold. If results are not clear – as Figure 4 suggests - this should be stated in the discussion under limitations.

Discussion

In the first paragraph the OR is discussed. Considering that quartile or tertile studies are included, the numerical value (1.5) should somewhat be de-emphasized although it is of considerable dimension.

When the authors allude to Mendelian randomization studies, they should go more into details which functional properties are attributable to the SNP`s. This text segment is too superficial. The landmark paper by B. Ference on LPL is not even considered (vide infra). Here, more work is necessary.

Line 229: more insight into fasting and postprandial triglyceride handling and the link to sd particles is necessary.

The discussion of guidelines is partly wrong. One would like to read more about levels of evidence and classes of recommendation, e.g. in the 2019 EAS/ESC guidelines. A more cautious statement is necessary.

References

Ref 55 is (Brian Ference et al.) is not referred to in the text, at least I did not find it, not even by a search program. This is unacceptable.

Reviewer #2: The authors report on the results of a systematic review and meta-analysis on the association of small dense LDL and CHD. The topic is of high interest and the manuscript provides interesting results on the question whether small dense LDL has a prognostic value for cardiovascular risk prediction. The manuscript is well written, statistical analyses are state of the art and the presentation of the results is appropriate. However, there are concerns about the search strategy and the discussion.

Specific comments:

1. The title is misleading. The review includes not only studies with sdLDL-cholesterol but also studies with sdLDL particle number.

2. The authors searched for "small dense LDL" or "small dense LDL-Cholesterol". However, some methods, such as NMR, determine the particle number and size and not the density. Thus, papers that use the term "small LDL" are likely to be overlooked. The authors should include also include the terms "dense LDL" and "dense LDL cholesterol" in their search strategy.

3. The most interesting question is whether sdLDL (particle number and / or cholesterol concentration) has a prognostic value in addition to established lipid biomarkers, especially LDL-C. In this context the adjustment strategies of the reviewed studies are important: some studies adjusted for lipid markers and others not.

Interestingly, studies that determined sdLDL particles and adjusted for lipid parameters (Zelijkovic, Kwon) reported ORs of 2.87 and 2.0, respectively, whereas the ORs for sdLDL-C plus adjustment for lipid parameters were lower (Arsenault, Koba) (Fig 2A and B). The authors should discuss this observation.

4. There are some inconsistencies regarding the methods between Table 1 and Supplemental table 4 (assay details): e.g. Mykannen (Ref 34) used electrophoresis, and the two studies of Koba from 2002 also (not LDL-EX from Denka).

5. The authors reviewed studies with incidence of CHD as primary outcome, studies using cardiovascular mortality as endpoint were not included. Therefore, the conclusions drawn from this meta-analysis on the prognostic value of sdLDL and its potential as a therapeutic target are limited to primary prevention.

The authors mentioned this in the methods section, but it should also be included in the limitations.

6. PLOS authors have the option to publish the peer review history of their article (what does this mean?). If published, this will include your full peer review and any attached files.

Reviewer #1: No

Reviewer #2: No

---

## [Author Response · Author response to Decision Letter 0]

14 Sep 2020

From this background, it is necessary

1) to more clearly state what is sdLDL(-C);

2) which particle characteristics are measured thereby; and

3) to more clearly state that the reports included investigated healthy of high risk individuals (which they do) but not patients with established disease (which they do not)

• Response: Done. We have more clearly stated which particle characteristics are measured and how. We added a paragraph (now paragraph 2 in the introduction) that explains how LDL is subfractionated and how sdLDL(-C) is defined. Within the paragraph under the heading “Study Eligibility Criteria”, we clarify that prospective studies investigated populations who did not have evidence of existing coronary artery disease at baseline or populations that had diabetes or HIV. Case control studies recruited controls who did not have evidence of existing coronary artery disease. A more detailed description of each patient population studied can be found in Table S4. 

SPECIFIC COMMENTS

The Introduction is appropriate to introduce the topic to the readership of an open access journal with holistic spectrum. However, one misses quotes on the pioneering work of Melissa Austin and the connection to triglyceride handling by Ron Krauss.

• Response: Done. We have added to our introduction quotes from Austin’s 1994 paper “Small, dense low-density lipoprotein as a risk factor for coronary heart disease”, Austin’s 2000 paper “Triglyceride, small, dense low-density lipoprotein, and the atherogenic lipoprotein phenotype”, and Krauss’s 1995 paper, “Dense Low-Density Lipoproteins and Coronary Artery Disease”. 

o Austin: “Austin has produced a large body of research further linking triglycerides and sdLDL as well as positing sdLDL as a risk factor for CHD, albeit based only on case control and cross-sectional studies.”

o Krauss: “They were initially described by Krauss to be associated with relative increases in plasma triglyceride and apolipoprotein B and posited to potentially underlie a familial predisposition to CHD.”

Methods

Again, it appears necessary to give more details on measurement of sd particles, also on their total cholesterol content, free and esterified cholesterol composition. Supplementary Table 4 is not sufficient for the reader who should learn about the validity and limitations of the assays in clinical terms

• Response: Done. We have included a brief exposition within our methods about how sdLDL-C and sdLDL were generally measured. Further details on how each individual study obtained their measurements as well as comments about the assay’s validity and limitations were added to Supplementary Table 5, which is referenced in the first paragraph of the results section. Details at the level of the type of cholesterol composition were not mentioned in any of the studies reviewed. We believe this is not necessarily a limitation of our study but rather a potential future direction in primary research to more granularly investigate types of cholesterol. 

Results

These are clearly stated. However, one misses more details on the absolute values of sd particle and their prospective value. One would like to know if there is a continuous risk or a threshold. If results are not clear – as Figure 4 suggests - this should be stated in the discussion under limitations.

• Response: Done. We have stated in the discussion that a continuous risk vs. a threshold risk could not be determined from the data available. Each of the individual three studies (Arai, Hoogeveen, and Tsai) reported significant trends of association across quantiles of sdLDL-C. However, from our study-level data, we were unable to conclusively determine whether there is a continuous or threshold risk, but so far, a continuous risk seems more plausible in the absence of evidence for a threshold risk. “Thus, this study is unable to determine whether there exists a dose-dependent risk or a threshold risk based on the concentration of sdLDL-C or sdLDL particles.”

Discussion

In the first paragraph the OR is discussed. Considering that quartile or tertile studies are included, the numerical value (1.5) should somewhat be de-emphasized although it is of considerable dimension.

• Response: Done. We reran the analyses after rerunning the search and included 4 additional articles in the meta-analysis. The pooled association for sdLDL studies is now 1.36 (95% CI: 1.21-1.52). We have de-emphasized the exact magnitude of the association by stating the range of the confidence interval. 

When the authors allude to Mendelian randomization studies, they should go more into details which functional properties are attributable to the SNP`s. This text segment is too superficial. The landmark paper by B. Ference on LPL is not even considered (vide infra). Here, more work is necessary.

• Response: Done. We have added more discussion about the functional properties attributable to SNPs associated with sdLDL-C as well as considered the work by Ference et. al. on LPL. “A landmark paper by Ference et. al. also demonstrated via MR that triglyceride-lowering lipoprotein lipase variants and low-density lipoprotein cholesterol-lowering variants were associated with lower risk of CHD per 10-mg/dL of apolipoprotein B-containing lipoproteins. Given the hypothesized mechanism between lipase and sdLDL formation, it will be interesting to similarly assess the association between sdLDL-lowering variants and risk of CHD

Line 229: more insight into fasting and postprandial triglyceride handling and the link to sd particles is necessary.

• Response: Done. We have added a discussion point regarding sdLDL levels in a postprandial state. “Specifically, with diet, an emerging field of research aims to understand the metabolomics of lipoproteins following a postprandial lipemia response to a meal, since standard fasting conditions prior to bloodwork are not necessarily representative of normal lifestyles. Although it has been hypothesized that hepatic lipase activity increases postprandially and results in the formation of more sdLDL particles, several studies have not been able to demonstrate a statistically significant increase in sdLDL levels.” 

The discussion of guidelines is partly wrong. One would like to read more about levels of evidence and classes of recommendation, e.g. in the 2019 EAS/ESC guidelines. A more cautious statement is necessary.

• Response: Done. We have rerun our search queries and identified 2 more randomized controlled trials which we have included in the meta-analysis. In brief, these RCTs have not only shown that lower sdLDL levels are associated with lower risk of CHD, but that also statins significantly reduce the levels of sdLDL particles. We believe this strengthens our main findings, as there is now causal evidence. We have included a substantial discussion on the implications of RCT evidence. Under the 2019 EAS/ESC guidelines, we conclude that given the newfound RCT evidence, more focus on studies looking at sdLDL-lowering interventions on incident CHD may be warranted. 

References

Ref 55 is (Brian Ference et al.) is not referred to in the text, at least I did not find it, not even by a search program. This is unacceptable.

• Response: By including discussion on Ference’s paper on LPL, the citation is now appropriate. 

Reviewer #2: The authors report on the results of a systematic review and meta-analysis on the association of small dense LDL and CHD. The topic is of high interest and the manuscript provides interesting results on the question whether small dense LDL has a prognostic value for cardiovascular risk prediction. The manuscript is well written, statistical analyses are state of the art and the presentation of the results is appropriate. However, there are concerns about the search strategy and the discussion.

• Response: We thank the reviewer for acknowledging the interest surrounding the topic and for their positive feedback on our writing, our statistical analyses and our presentation. 

Specific comments:

1. The title is misleading. The review includes not only studies with sdLDL-cholesterol but also studies with sdLDL particle number.

• Response: Done. We have changed the title to “Association of Small, Dense LDL-Cholesterol Concentration and Lipoprotein Particle Characteristics with Coronary Heart Disease: A Systematic Review and Meta-analysis”.

2. The authors searched for "small dense LDL" or "small dense LDL-Cholesterol". However, some methods, such as NMR, determine the particle number and size and not the density. Thus, papers that use the term "small LDL" are likely to be overlooked. The authors should include also include the terms "dense LDL" and "dense LDL cholesterol" in their search strategy.

• Response: Done. Repeating the search strategy by adding “small LDL”, “dense LDL” and “dense LDL cholesterol” (still limiting the date range to the original date of January 29, 2020) resulted in 4 additional articles included in the meta-analysis. The PRISMA flow diagram was remade and all primary and supplementary analyses were rerun with the additional data. Two randomized controlled trials were also identified, lending some causal evidence to our findings. Overall, the pooled measures of association were still significantly positive, albeit attenuated; however, the attenuation is likely a result of the high heterogeneity between studies. Thus, the core of our conclusions remains unchanged, and in fact, with the addition of more studies thanks to your suggestion, our pooled estimates are more precise, and we believe the addition of RCTs strengthens our argument. 

3. The most interesting question is whether sdLDL (particle number and / or cholesterol concentration) has a prognostic value in addition to established lipid biomarkers, especially LDL-C. In this context the adjustment strategies of the reviewed studies are important: some studies adjusted for lipid markers and others not.

Interestingly, studies that determined sdLDL particles and adjusted for lipid parameters (Zeljkovic, Kwon) reported ORs of 2.87 and 2.0, respectively, whereas the ORs for sdLDL-C plus adjustment for lipid parameters were lower (Arsenault, Koba) (Fig 2A and B). The authors should discuss this observation.

• Response: Done. We would like to clarify that, according to Figure 2, the studies that determined sdLDL particles and adjusted for lipid parameters were Kwon and Bucher (ORs of 2.0 and 1.0 respectively) whereas the studies that determined sdLDL-C particles and adjusted for lipid parameters were Tsai, Hoogeveen, and Arsenault (ORs of 2.41, 1.51, and 1.39 respectively). In Figure S1, which shows forest plots for sdLDL and sdLDL-C studies subgrouped by adjustment level, there seems to be no apparent differences in the pooled measures of association across adjustment levels. That being said, we agree that adjusting for lipid markers is important, and in fact, the pooled measure of association adjusting for lipid markers no longer retained significance (1.54, 95% CI: 0.97, 2.43). We have included this observation in our discussion.

• Furthermore, we had an idea to compare the LDL-adjusted estimates with the non-LDL-adjusted estimates in studies that adjusted for LDL (Arsenault, Mackey, Koba 2008, Kwon, and Williams); however, these studies did not report the non-LDL-adjusted estimates so we could not evaluate whether or not the addition of LDL attenuated the association between sdLDL and incident CHD. 

4. There are some inconsistencies regarding the methods between Table 1 and Supplemental table 4 (assay details): e.g. Mykannen (Ref 34) used electrophoresis, and the two studies of Koba from 2002 also (not LDL-EX from Denka).

• Response: Done. We thank the reviewer for catching this error and have fixed it.

5. The authors reviewed studies with incidence of CHD as primary outcome, studies using cardiovascular mortality as endpoint were not included. Therefore, the conclusions drawn from this meta-analysis on the prognostic value of sdLDL and its potential as a therapeutic target are limited to primary prevention.

The authors mentioned this in the methods section, but it should also be included in the limitations.

• Response: Done. Most of the studies that we included did include fatal events in their endpoint definitions. We have clarified this in the methods by stating that the outcome of interest includes fatal events as well. However, we acknowledge that we did not include studies in which patients who experienced fatal events due to heart failure or other cardiovascular outcomes. We include the following statement in our limitations: “Finally, we did not include studies in which patients who experience fatal events due to heart failure or other cardiovascular outcomes, which limits the conclusions drawn from this study on the prognostic value of sdLDL and sdLDL-C to primary prevention.”

---

## [Decision Letter · Decision Letter 1]

26 Oct 2020

Association of Small, Dense LDL-Cholesterol Concentration and Lipoprotein Particle Characteristics with Coronary Heart Disease: A Systematic Review and Meta-analysis

PONE-D-20-14231R1

Dear Dr. Liou,

We’re pleased to inform you that your manuscript has been judged scientifically suitable for publication and will be formally accepted for publication once it meets all outstanding technical requirements.

Kind regards,

Andreas Zirlik, MD

Academic Editor

PLOS ONE

Additional Editor Comments (optional):

Reviewers' comments:

Reviewer's Responses to Questions

**Comments to the Author**

1. If the authors have adequately addressed your comments raised in a previous round of review and you feel that this manuscript is now acceptable for publication, you may indicate that here to bypass the “Comments to the Author” section, enter your conflict of interest statement in the “Confidential to Editor” section, and submit your "Accept" recommendation.

Reviewer #1: All comments have been addressed

Reviewer #2: All comments have been addressed

2. Is the manuscript technically sound, and do the data support the conclusions?

Reviewer #1: Yes

Reviewer #2: Yes

3. Has the statistical analysis been performed appropriately and rigorously? 

Reviewer #1: Yes

Reviewer #2: Yes

4. Have the authors made all data underlying the findings in their manuscript fully available?

Reviewer #1: Yes

Reviewer #2: Yes

5. Is the manuscript presented in an intelligible fashion and written in standard English?

Reviewer #1: Yes

Reviewer #2: Yes

6. Review Comments to the Author

Reviewer #1: All points are well addressed now. The authors have done a very good Job. They have included more references, addressed all criticisms and even made further calculations.

Reviewer #2: (No Response)

7. PLOS authors have the option to publish the peer review history of their article (what does this mean?). If published, this will include your full peer review and any attached files.

Reviewer #1: **Yes: **Heinz Drexel, MD, FESC, FAHA, FRCP (Ed.), Professor of Medicine

Reviewer #2: No

---

## [Editor Report · Acceptance letter]

28 Oct 2020

PONE-D-20-14231R1 

Association of Small, Dense LDL-Cholesterol Concentration and Lipoprotein Particle Characteristics with Coronary Heart Disease: A Systematic Review and Meta-analysis 

Dear Dr. Liou:

I'm pleased to inform you that your manuscript has been deemed suitable for publication in PLOS ONE. Congratulations! Your manuscript is now with our production department. 

Kind regards, 

on behalf of

Univ. Prof. Dr. Andreas Zirlik 

Academic Editor

PLOS ONE